# Prevention of Inflammation-Driven Colon Carcinogenesis in Human MUC1 Transgenic Mice by Vaccination with *MUC1* DNA and Dendritic Cells

**DOI:** 10.3390/cancers15061920

**Published:** 2023-03-22

**Authors:** Retno Murwanti, Kaori Denda-Nagai, Daisuke Sugiura, Kaoru Mogushi, Sandra J. Gendler, Tatsuro Irimura

**Affiliations:** 1Laboratory of Cancer Biology and Molecular Immunology, Graduate School of Pharmaceutical Sciences, The University of Tokyo, 7-3-1 Hongo, Bunkyo-ku, Tokyo 113-0033, Japan; 2Faculty of Pharmacy, Universitas Gadjah Mada, Sekip Utara, Yogyakarta 55283, Indonesia; 3Intractable Disease Research Center, Graduate School of Medicine, Juntendo University, 2-1-1 Hongo, Bunkyo-ku, Tokyo 113-8421, Japan; 4Laboratory of Molecular Immunology, Institute for Quantitative Biosciences, The University of Tokyo, Yayoi, Bunkyo-ku, Tokyo 113-0032, Japan; 5Department of Immunology, Mayo Clinic Arizona, 13400 E. Shea Blvd., Scottsdale, AZ 85259, USA; 6Division of Glycobiologics, Graduate School of Medicine, Juntendo University, 2-1-1 Hongo, Bunkyo-ku, Tokyo 113-8421, Japan

**Keywords:** colon carcinogenesis, dendritic cells, DNA vaccination, human MUC1 transgenic mice

## Abstract

**Simple Summary:**

Colorectal cancer is one of the most common cancers worldwide and improved treatments are needed. The present research focused on the use of a DNA vaccine targeted at one of the commonly found tumor-associated antigens, human MUC1. MUC1 transgenic mice were vaccinated with *MUC1* DNA mixed with bone-marrow-derived dendritic cells (BMDCs). Then, inflammation-driven colon cancer was induced by injection of a carcinogen, azoxymethane, and administration of dextran-sulfate-sodium-containing drinking water. This mouse model of colon cancer resembles the colon cancer found in humans. The number and size of colonic tumors were compared to mice vaccinated with *MUC1* DNA alone or BMDCs alone. The combination of *MUC1* DNA and BMDCs reduced the tumor burden more than in the controls, showing an additive preventive effect. These findings demonstrate that in this human MUC1 transgenic mouse model, MUC1-expressing colonic tumors can be partially prevented by a vaccine containing *MUC1* DNA and BMDCs.

**Abstract:**

The preventive efficacy of *MUC1*-specific DNA immunization on inflammation-driven colon carcinogenesis in human MUC1 transgenic (MUC1.Tg) mice was investigated. Mice were vaccinated with *MUC1* DNA mixed with autologous bone-marrow-derived dendritic cells (BMDCs), and then colonic tumors were induced by azoxymethane (AOM) injection and oral administration of dextran sulfate sodium (DSS). Two types of tumors, squamous metaplasia and tubular adenoma, were observed. Both expressed high levels of MUC1 as indicated by the binding of anti-MUC1 antibodies with different specificities, whereas MUC1 expression was not detected in normal colonic mucosa. When mice were immunized with *MUC1* DNA + BMDCs, tumor incidence, tumor number, and tumor size were significantly reduced. In contrast, vaccination with *MUC1* DNA alone or BMDCs alone was ineffective in reducing tumor burden. Inflammation caused by DSS was not suppressed by the *MUC1* DNA + BMDCs vaccination. Furthermore, MUC1 protein expression levels, as judged by anti-MUC1 antibody binding in tumors grown after vaccination, did not significantly differ from the control. In conclusion, an inflammation-driven carcinogenesis model was established in MUC1.Tg mice, closely resembling human colon carcinogenesis. In this model, vaccination with *MUC1* DNA + BMDCs was effective in overriding MUC1 tolerance and reducing the tumor burden by a mechanism not affecting the level of colonic inflammation.

## 1. Introduction

Colorectal cancer is one of the most common cancers worldwide, leading to significant morbidity and mortality [1]. Patients with ulcerative colitis and Crohn’s disease are at an increased risk for developing colorectal cancer. Although the genetic basis of colorectal cancer predisposition in these inflammatory bowel diseases has not been clearly elucidated, it is believed that chronic inflammation is one of the pathogenic sequences of colorectal carcinogenesis. The connection between inflammation and carcinogenesis is well established and has received a great body of supporting evidence from genetic, pharmacological, and epidemiological data [2]. 

Despite efforts to improve conventional therapy, the mortality of colorectal cancer is still high, particularly related to the recurrence of the disease and the occurrence of liver metastases. Novel approaches, such as cancer immunotherapy targeting tumor-associated antigens, which are expected to show fewer side-effects and to provide better prevention of metastasis and recurrence than conventional cancer therapy, represent an attractive therapeutic avenue. 

There has been great interest in cancer vaccines, which have the potential for controlling metastatic disease, prolonging the time to recurrence, and ultimately serving as a preventive measure. DNA vaccines have several advantages over other vaccine technologies [3]. DNA vaccines have been reported to be beneficial for maintaining high levels of tumor antigen expression at the vaccination site and for eliciting strong antitumor immunity in hosts [4,5,6] Several studies in experimental mouse models showed that tumor growth was suppressed successfully by vaccination with DNA encoding tumor antigens [7,8,9,10,11,12]. In light of the recent progress regarding the use of mRNA and DNA vaccination technology to stem the COVID-19 pandemic [13,14] and to provide new treatment possibilities for cancer [15], genetic vaccines may soon become an accepted therapeutic modality.

In the present study, we focused on MUC1 as a therapeutic target. MUC1 is a tumor-associated transmembrane antigen expressed at low levels on the apical surface of ductal epithelial cells in several organs, and overexpressed in the majority of adenocarcinomas and their precursor lesion, where its tandem repeat region is reported to have truncated *O*-glycans [16,17,18,19,20]. As a result, truncated novel glycopeptide epitopes are processed and presented to the immune system [10,21,22,23]. MUC1 localization in tumors has not only been reported to be restricted to the apical surface, but has also been found all around the cell surface and in the cytoplasm. MUC1’s highly increased expression and altered glycosylation in tumors provide a source of predictable disease-associated neo-epitopes that could exert an important influence on the innate immune system and represent targets of adaptive immunity [24,25].

Tumor-associated MUC1 has long been considered as a potential target molecule that might be able to elicit an immune response that then provides protection against MUC1-expressing tumor cells. Various strategies targeting the MUC1 tumor antigen to prevent, stabilize, or eradicate cancerous lesions have been pursued [26,27]. MUC1 peptides [28], dendritic cells (DCs) transfected with *MUC1* cDNA [29,30], MUC1-mannan and MUC1-mannose fusion proteins [31,32], and *MUC1* plasmid DNA [33,34] have all been reported to have the ability to suppress the growth of MUC1-expressing tumors in wild-type mice. In addition, studies in human MUC1-transgenic (MUC1.Tg) mice have shown that vaccination with MUC1 peptides [35,36] and *MUC1* DNA [37,38] can effectively break the tolerance to injected MUC1-expressing tumor cells. In humans, injection of oxidized mannan-MUC1 after breast cancer surgery successfully prevented tumor recurrence for up to 15 years [39,40]. 

We have previously shown that a *MUC1* DNA vaccine can prevent colonic tumors and lung metastasis in C57BL/6 mice [41,42]. In the present study, we performed experiments using MUC1.Tg mice, which express the full-length human MUC1 under its own promoter, thus having a similar MUC1 expression level and pattern as seen in humans [43]. In contrast to previous studies that injected MUC1-expressing tumor cells to induce cancer, here, we optimized the azoxymethane-dextran sulfate sodium (AOM-DSS) model [44,45,46] in MUC1.Tg mice to induce colorectal carcinogenesis. By using this model, we showed the development of MUC1-expressing colorectal tumors in an environment and immune system that recognizes MUC1 as a self (de novo) antigen and closely mimics the colitis-associated colorectal carcinogenesis conditions as seen in humans. Further, we sought to investigate if the anti-tumor immunity can be improved by addition of dendritic cells (DCs). DCs have been reported as professional antigen-presenting cells that play a critical role in the induction and regulation of immune responses [47]. It has been proposed that the manipulation of DCs as a “natural” vaccine adjuvant may prove to be a particularly effective way to stimulate antitumor immunity [48,49].

Here, we demonstrated the preventive efficacy of the *MUC1* DNA vaccine when combined with purified autologous bone-marrow-derived DCs (BMDCs) against a colitis-associated colorectal tumor in MUC1.Tg mice. Administration of the *MUC1* DNA vaccine alone or BMDCs alone was ineffective. These findings highlight the importance of combining the DNA vaccine with dendritic cells to achieve immune protection against tumor formation. 

## 2. Materials and Methods

### 2.1. Animals

MUC1.Tg mice with a C57BL/6 background [43] were maintained in The University of Tokyo, Graduate School of Pharmaceutical Sciences’ animal facility under specific-pathogen-free conditions. MUC1-positive transgenic mice were confirmed by genotyping PCR as previously described [43]. Male MUC1.Tg mice aged 6 to 8 weeks were used. All experiments were approved by the Bioscience Committee of the Graduate School of Pharmaceutical Sciences of The University of Tokyo (approval code: PH30-4) and performed according to the guidelines of the Bioscience Committee of The University of Tokyo.

### 2.2. Antibodies

MUC1-specific Abs were biotin-conjugated chicken anti-MUC1 cytoplasmic tail (CTP) polyclonal antibody (pAb) kindly provided by Dr. Veer P. Bhavanandan (retired from the Pennsylvania State University, Old Main, State College, PA, USA) and biotin-conjugated monoclonal antibody (mAb) MY1E.12, biotin-conjugated mAb HMFG1, as well as biotin-conjugated mAb HMFG2 kindly provided by Dr. Joyce Taylor-Papadimitriou (Guys Hospital, London, UK). Biotin-conjugated IgG2a (BioLegend, Inc., San Diego, CA, USA) and biotin-conjugated IgY (Santa Cruz Biotechnology, Santa Cruz, CA, USA) were used as isotype controls.

### 2.3. Construction of MUC1 cDNA 

Full-length human *MUC1* cDNA containing 22 tandem repeats was provided originally in the pDKOF vector by Dr. Olivera J. Finn (University of Pittsburgh, Pittsburgh, PA, USA) [50]. From the pDKOF vector, the cDNA was cut out and re-cloned into the HindIII site of the pcDNA3.1 vector (Invitrogen; Thermo Fisher Scientific, Waltham, MA, USA), which includes the CMV promoter (pcDNA 3.1.-MUC1). A plasmid without *MUC1* cDNA (pcDNA3.1) was used as the control. *MUC1* cDNA in the pcDNA3.1 vector (pcDNA-MUC1) and pcDNA3.1 plasmid alone were amplified in the *Escherichia coli* strain DH5 and purified using Qiagen Mega-Plasmid columns (Qiagen, Inc., Germantown, MD, USA) according to the manufacturer’s protocol. This purification kit ensures a residual endotoxin level of 9.3 EU/µg DNA. Purification of DNA plasmid was confirmed by gel electrophoresis. The DNA was dissolved in Hanks’ Balanced Salt Solution (HBSS) at a concentration of 4 mg/mL and stored at −20 °C.

### 2.4. Preparation of BMDCs

BMDCs were prepared from the femur and tibia of MUC1.Tg mice as previously described [51] with the following modifications. Cells (1 × 10^6^ cells/mL) were cultured in RPMI1640 medium supplemented with 10% FCS, penicillin and streptomycin (1000 IU/mL and 10 mg/mL, respectively), 2 mM glutamine, and 0.05 mM 2-mercaptoethanol in the presence of 1000 U/mL of recombinant murine GM-CSF. On day 6, CD11c^+^ cells were purified using anti-CD11c MicroBeads and AutoMACS (Miltenyi Biotec, Bergisch Gladbach, Germany) and re-plated into fresh 24-well plates. On day 7, non-adherent cells were collected and suspended in 50 µL of HBSS. Cell viability as checked by the trypan blue exclusion test was routinely higher than 90%. The maturation level was checked by staining with anti-CD11c and anti-MHCII antibodies and flowcytometric analysis. Cells consisted of MHCII^low-intermediate^ (50%) and MHCII^hi^ (20–30%) CD11c^+^ BMDCs. These cells were used for vaccination. 

### 2.5. Vaccination 

Mice were randomly allocated to the treatment groups and received three vaccinations at weekly intervals in the footpad with 100 μg of *MUC1* DNA dissolved in 50 μL of HBSS mixed with or without 5 × 10^5^ BMDCs per mouse, per vaccination.

### 2.6. Experimental Colitis-Associated Colorectal Carcinogenesis

A single dose of 10 mg/kg body weight AOM (Sigma-Aldrich; Merck, Darmstadt, Germany), dissolved in sterile saline before use, was injected intraperitoneally (i.p.) 7 days after the last vaccination. Administration of three cycles of 1% (*w*/*v*) DSS (MP Biomedicals, LLC, Aurora, OH, USA) in drinking water was started 7 days after AOM injection for 4 days in the first cycle, and 5 days in the second and third cycle, followed by regular drinking water in between. Mice who died during the DSS treatment were not included into the data analysis. Mice were sacrificed 2 weeks after the last DSS exposure. The experimental protocol is schematically shown in Figure 1A. Colon samples were embedded in Tissue Tek O.C.T. Compound (Sakura Finetek, Tokyo, Japan). 

### 2.7. Gross and Histopathological Examination

General observation was conducted to assess clinical symptoms and mortality during vaccinations and AOM-DSS treatment. Tumor size was measured by measuring the length and width of each tumor with a ruler. Tumor numbers were counted under a microscope. Histopathological alterations in the colon were examined using H&E-stained (Sakura Finetek) 6 μm thick serial cryosections of O.C.T. Compound-embedded colon samples. Colonic tumors were classified according to Boivin [52] and Reeve [53]. 

### 2.8. Immunohistochemical Analysis

For MUC1 detection, 6 μm thick cryostat sections of normal colon as well as colonic tissue with tubular adenoma or squamous metaplasia were used. Sections were pretreated for 30 min with 0.3% hydrogen peroxide at room temperature to eliminate endogenous peroxidase and washed three times with Dulbecco’s PBS. All endogenous biotin, biotin receptors, or avidin binding sites were blocked using the VECTASTAIN ABC HRP Kit (Vector Laboratories, Burlingame, CA, USA), according to the manufacturer’s protocol. The slides were incubated overnight at 4 °C with biotin-conjugated primary Abs or isotype control Abs at 5 μg/mL diluted in 3% BSA in PBS. Subsequently, Streptavidin Horseradish Peroxidase Conjugate (Invitrogen) was used at 1/200 dilution. Sections were counterstained with H&E. Immunoreactivity against MUC1 Ab was assessed under a microscope (Leica Microsystems, Wetzlar, Germany). To measure MUC1-glycoform expression in tumors, semi-quantitative analysis of immunohistochemically stained sections was performed by roughly estimating the percentage of MUC1-positive cells as compared to all epithelial cells.

### 2.9. Effect of MUC1 DNA Vaccination on Azoxymethane-Dextran Sulfate Sodium (AOM-DSS)-Induced Colorectal Inflammation

Seven days after the last vaccination, 10 mg/kg body weight of AOM diluted in sterile saline was injected i.p. Administration of 1% (*w*/*v*) DSS diluted in drinking water was started on day 7 after AOM injection for 1 cycle of 5 days. Mice were sacrificed on day 5 of DSS administration, and the body weight, hemoccult and/or presence of blood, and histological features were analyzed. The experimental setup is depicted in Figure 4A. Body weight change was determined by measuring the body weight on Day 1 and Day 5 of DSS treatment, the time period that had shown the largest body weight loss in a preliminary experiment. Bleeding scores were determined as 0 = no bleeding; 1 = positive Guaiac occult blood test (minimal color change to green); 2 = positive Guaiac occult blood test (maximal color change to blue); 3 = blood visibly present in the stool with no clotting on the anus; 4 = gross bleeding from the anus with clotting present. Inflammation with or without mucosal ulcers in the large bowel was scored on H&E-stained sections. For scoring, large intestinal inflammation was graded according to the following morphological criteria: grade 0 = normal appearance; grade 1 = shortening and loss of the basal 1/3 of the actual crypts with mild inflammation in the mucosa; grade 2 = loss of the basal 2/3 of the crypts with moderate inflammation in the mucosa; grade 3 = loss of entire crypts with severe inflammation in the mucosa and submucosa, but retention of the surface epithelium; grade 4 = presence of mucosal ulcers with severe inflammation (infiltration of neutrophils, lymphocytes, and plasma cells) in the mucosa, submucosa, muscularis propria, and/or subserosa. Two sections of the colon (one colorectal and one anal) were assessed and scored for each mouse and the inflammatory score was expressed as the mean of these scores.

### 2.10. Statistical Analysis

To analyze differences in tumor numbers and MUC1-expressing cells, the Steel–Dwass test was performed using R [54]. To analyze differences in AOM-DSS-induced intestinal inflammation, the non-parametric Mann–Whitney U test was performed using GraphPad Prism version 6.00 [55]. Results were considered significant at *p* < 0.05.

## 3. Results

### 3.1. MUC1 Expression in Tumors Developed by Colitis-Associated Colorectal Carcinogenesis in MUC1.Tg Mice 

The experimental protocol is schematically shown in Figure 1A. AOM-DSS treatment induced the growth of tumor polyps mainly in the distal part of the colon as shown in Figure 1B–D. Figure 1E shows a normal part of the colon.

**Figure 1 cancers-15-01920-f001:**
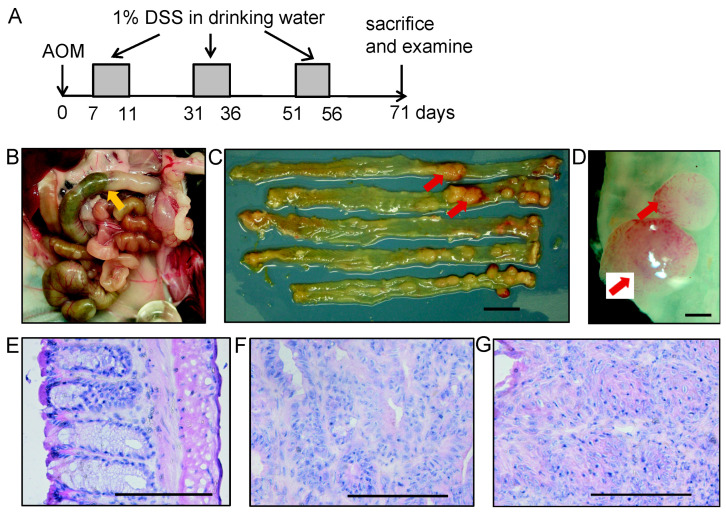
Gross pathology and histopathology of colitis-associated colorectal tumor induced by azoxymethane-dextran sulfate sodium (AOM-DSS) in MUC1.Tg mice. (**A**) Experimental protocol. Azoxymethane (AOM) was injected at 10 mg/kg body weight in the intraperitoneal site, followed by periodical administration of 1% (*w*/*v*) dextran sulfate sodium (DSS) in drinking water. Mice were sacrificed two weeks after the last DSS administration. (**B**) Colon position in the mouse abdomen as indicated by the yellow arrow. (**C**,**D**) Tumors developed as pedunculated nodules observed in the distal part of the colon as indicated by red arrows. (**C**) Gross pathological appearance of colorectal tumors showed pale to reddish polypoid nodules in the colorectal site. Histopathological analysis was performed on H&E-stained sections. Representative histopathological images of (**E**) normal colon, (**F**) tubular adenoma, and (**G**) squamous metaplasia of AOM-DSS-treated MUC1.Tg mice. Images in (**E**–**G**) were captured at ×200 magnification. Scale bars show (**C**) 10 mm, (**D**) 1 mm, and (**E**–**G**) 100 μm.

Some tumors were tubular adenomas with various grades of dysplasia (Figure 1F). Another type of tumor observed was squamous metaplasia with atypia, which was characterized by a stratified squamous epithelium that developed in the colorectal area (Figure 1G). 

MUC1 expression has been reported in human colon adenoma and adenocarcinoma, but not in normal colonic epithelia [16,56]. We hypothesized that AOM-DSS treatment could induce MUC1-expressing colorectal tumor formation in MUC1.Tg mice. Therefore, we performed immunohistochemical analysis in non-vaccinated MUC1.Tg mice to confirm MUC1 expression in AOM-DSS-induced colorectal tumors by using anti-MUC1 antibodies (Abs) with different specificities. Anti-MUC1 CTP Ab recognizes the MUC1 cytoplasmic tail regardless of glycosylation status [57]. Monoclonal Ab MY.1E12 recognizes sialyl-T MUC1 [58,59]. The binding of HMFG1 mAb to the PDTR epitope in MUC1 is partially influenced by certain nearby glycans [59], while the binding of HMFG2 mAb to MUC1 is glycosylation-independent [59]. The results showed that various MUC1 glycoforms were expressed in AOM-DSS induced tumors (Figure 2A). MUC1 was expressed mostly along the apical membrane of adenomatous cells (Figure 2A,f–h). However, in squamous metaplasia, MUC1 was highly expressed, not only in the apical part of tumor cells but also on the whole cell surface and in the cytoplasm (Figure 2A,k–m). The staining intensity of all anti-MUC1 Abs was stronger in squamous metaplasia tumors than in colorectal adenoma tumors (Figure 2A–E). 

### 3.2. Preventive Efficacy of MUC1 DNA Vaccine on Experimental Colitis-Associated Colorectal Carcinogenesis

To evaluate the efficacy of MUC1-targeted immunotherapy, we examined the preventive effect of the *MUC1* DNA vaccine in an experimental colitis-associated colorectal carcinogenesis model. MUC1.Tg mice were divided into five groups (*n* = 10 per group) and vaccinated with the following: (1) mock plasmid (pcDNA), (2) *MUC1* plasmid (MUC1), (3) BMDCs (BMDCs), (4) pcDNA mixed with BMDCs (pcDNA + BMDCs), and (5) MUC1 + BMDCs, each followed by AOM-DSS treatment. The experimental protocol is schematically shown in Figure 3A.

Vaccinated mice were observed for 2 weeks after the last DSS treatment. Regarding the general observation, no differences in clinical symptoms, body weight, or length of the colon were observed between groups. Severe bleeding was observed during or after DSS administration in all groups. To judge whether the vaccination inhibits tumor formation, tumor multiplicity was evaluated in each treatment group. The results showed that although colorectal tumor formation could not be prevented completely, vaccination with MUC1 + BMDCs reduced the tumor incidence by 37.5% compared to the control (Figure 3B). The tumor multiplicity in MUC1.Tg mice vaccinated with MUC1 + BMDCs was significantly lower compared to mice vaccinated with BMDCs (*p* < 0.05) and mice vaccinated with pcDNA + BMDCs (*p* < 0.05) (Figure 3C). In addition, the tumor size distribution showed that in MUC1.Tg mice vaccinated with MUC1 + BMDCs, the frequency of smaller tumors (1 mm^2^) was highest, while that of larger tumors (ranging from 3 mm^2^ to 54 mm^2^) was lowest compared to all other groups (Figure 3D). These results strongly indicate that combining *MUC1* DNA and BMDCs is critical to inducing an effective prevention of tumor formation in the colitis-associated colorectal carcinogenesis model. 

### 3.3. Effect of MUC1 DNA Vaccination on AOM-DSS Induced Colorectal Inflammation

In the present study, we used a carcinogenesis model in which inflammation is crucial to promoting tumorigenesis. Hence, we evaluated the effect of vaccination on inflammation induced by AOM-DSS administration (Figure 4A). The results showed that there were no significant differences regarding body weight loss (Figure 4B) and bleeding scores (Figure 4C). Colorectal and anal regions of both groups of mice showed a loss of entire crypts with severe inflammation in the mucosa and submucosa (Figure 4D). Inflammation scores between mice vaccinated with pcDNA + BMDCs and mice vaccinated with MUC1 + BMDCs were not different (Figure 4E). These results clearly show that there was no significant difference in the tissue damage due to DSS administrations between mice vaccinated with pcDNA + BMDCs or MUC1 + BMDCs.

## 4. Discussion

This study was designed to evaluate the efficacy of *MUC1* DNA vaccination in the prevention of colitis-associated colorectal carcinogenesis. Our results strongly suggest that the MUC1 vaccine is effective in preventing, although not completely, colitis-associated colorectal tumor formation when combined with purified CD11c^+^ BMDCs. This study is the first to report the efficacy of a combined MUC1 + BMDC vaccine in the prevention of spontaneous, carcinogen-induced colorectal carcinogenesis in MUC1.Tg mice. 

While several previous studies have examined the efficacy of *MUC1* DNA vaccination in cancer, these studies were mostly conducted in wild-type mice and/or administered exogenous tumor cells. The significance of the present study lies in (1) the use of MUC1.Tg mice who express de novo MUC1 as a self-antigen and have been reported to be tolerant to MUC1 [43], (2) the use of an inflammation-induced colon cancer model that closely resembles the colon carcinogenesis process seen in humans, and (3) the use of DCs to enhance the anti-tumor immune response. Previously, we have shown that the growth of administered MUC1-expressing tumor cells in MUC1.Tg mice is enhanced, likely due to the presence of MUC1-specific regulatory T cells [60]. The results of the present study show that by the combination of *MUC1* DNA and BMDCs, an immune response could be elicited to eliminate MUC1-expressing tumor cells at an early stage of carcinogenesis, providing evidence that MUC1 tolerance can be overcome in this clinically relevant colon cancer model.

The MUC1.Tg AOM-DSS carcinogenesis model provides high reproducibility and a simple mode of application and resembles human colorectal tumors in pathological and molecular aspects. Therefore, this model provides new opportunities to develop MUC1-targeted cancer immunotherapy for humans. MUC1 has been reported to be highly expressed in human colorectal adenoma and adenocarcinoma, particularly at the advanced stage [16]. Therefore, this vaccine might also be effective in the prevention of colorectal cancer recurrence.

In the present study, we showed that vaccination with MUC1 + BMDCs significantly reduced tumor incidence and tumor burden by inhibiting colitis-associated colorectal tumor formation. In addition, this combined vaccine might also inhibit colorectal tumor growth, represented by the smaller tumor size in this group. In contrast, vaccination using MUC1 or BMDCs alone was ineffective. Vaccination with DNA-encoding tumor antigens has been reported to be beneficial for maintaining high levels of tumor antigen expression at the vaccination site and for inducing strong antitumor immunity [61,62]. Regarding MUC1, this hypothesis has been supported by preclinical studies in mice. For instance, Kontani et al. reported that injecting *MUC1* DNA and non-primed DCs at the same site, in contrast to injecting at different sites, enhanced MUC1-specific antitumor immunity in wild-type C57BL/6 mice [63]. Their results showed that inoculated DCs were able to process and express *MUC1* DNA-encoding antigens at the inoculation site.

Our findings showed that, although this vaccine strategy successfully reduced tumor formation, complete inhibition was not achieved. It has been reported that tumors can escape from the immune system by reducing their tumor-associated antigen [64]. Although the glycoform of MUC1 to which the strong immune response seen in the present study was elicited is not known, the tumors in MUC1.Tg mice, which became a target of the immune response, appeared to express a variety of glycoforms. Further, immunohistochemical analysis of DSS-induced tumors using MY.1E12 mAb revealed that the types of tumors observed showed expression of MUC1 irrespective of whether the mice had been vaccinated or not. These preliminary results suggest that downregulation of MUC1 antigen levels may not be the mechanism used by these tumors to escape from MUC1-specific immune responses. The combination with other therapies should be one strategy to improve the tumor suppression efficacy.

A previous study, which used interleukin 10-deficient mice crossed with MUC1.Tg mice that developed spontaneous inflammatory bowel disease, showed that vaccination with a Tn-MUC1 100-mer peptide could reduce inflammation [24]. In our model, DSS-induced chronic inflammation is critical to the induction of colorectal tumor formation. Therefore, we asked whether the effects of the MUC1 + BMDC vaccine were mediated by an inhibition of inflammation. The results showed that there was no significant difference in the tissue damage due to DSS administrations between pcDNA + BMDCs- and MUC1 + BMDCs-vaccinated mice, suggesting that the tumor suppression effects of the MUC1 + BMDC vaccine in the AOM-DSS model are not mediated by inhibition of inflammation. Taken together with the fact that tumor suppression was only seen when BMDCs were administered together with *MUC1* DNA, it is reasonable to assume that a MUC1-specific acquired immune response is involved. Previously, we vaccinated wild-type mice with *MUC1* DNA and challenged them by injection of MUC1-expressing tumor cells in an experimental model of colon carcinoma. In that model, CD4^+^ T cells were found to be responsible for the anti-tumor response both at the orthotopic and the metastatic tumor site [42]. What mechanisms mediate the effect of tumor prevention by vaccination with MUC1 + BMDCs in the AOM-DSS-induced colorectal carcinogenesis model remains to be elucidated. 

## 5. Conclusions

We have reported that vaccination with *MUC1* DNA combined with BMDCs reduced the tumor burden in colitis-associated colorectal carcinogenesis in MUC1.Tg mice. The vaccination inhibited not only tumor formation but also tumor growth. This study highlights the crucial contribution of BMDCs to enhancing antitumor immunity of the *MUC1* DNA vaccine and provides supporting preclinical evidence for its tolerance-breaking efficacy in a clinically relevant mouse model of human colon cancer.

## Figures and Tables

**Figure 2 cancers-15-01920-f002:**
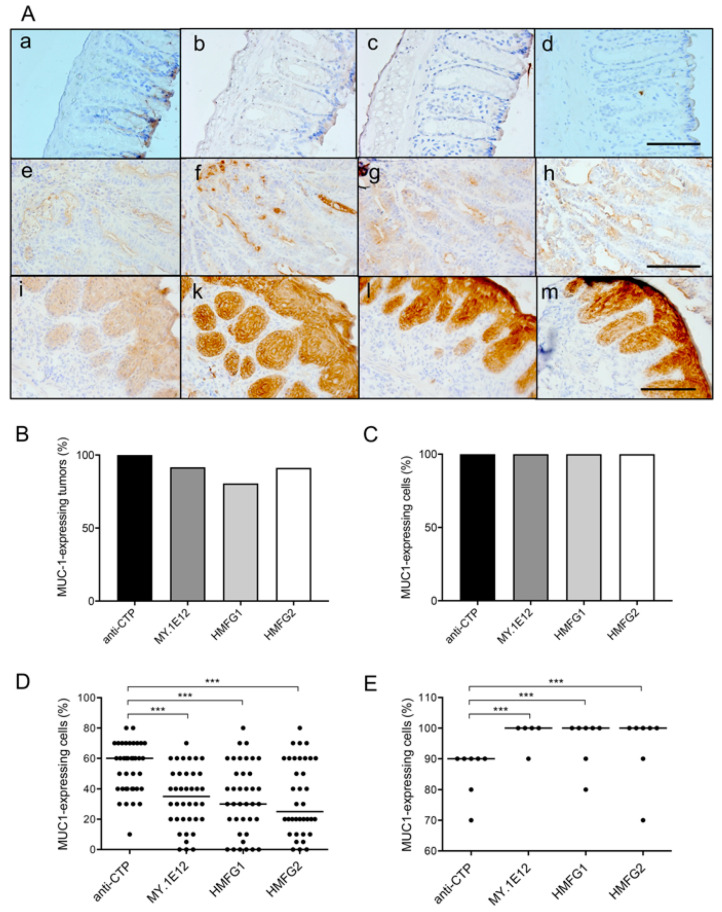
MUC1 staining profiles and percentage of MUC1-expressing tumors and cells in colorectal tumors induced by azoxymethane-dextran sulfate sodium treatment in non-vaccinated MUC1.Tg mice as investigated using four MUC1-specific antibodies. (**A**) Representative images from immunohistochemistry (IHC) using four anti-MUC1 antibodies (Abs) with different specificities. The following antibodies were used: polyclonal Ab (pAb) anti-CTP (**a**,**e**,**i**), monoclonal Ab (mAb) MY.1E12 (**b**,**f**,**k**), mAb HMFG1 (**c**,**g**,**l**), and mAb HMFG2 (**d**,**h**,**m**). Normal colon of MUC1.Tg mice sections showed negative binding with all anti-MUC1 Abs (upper panel). Bindings of various anti-MUC1 Abs were observed in tubular adenoma (*n =* 36, **f**–**h**) and squamous metaplasia (*n* = 10, **k**–**m**). All images were captured at ×200 magnification. Scale bar shows 100 μm. Semi-quantitative analysis of IHC staining results was performed to evaluate MUC1 glycoform expression in tumors. (**B**,**C**) Percentage of MUC1 glycoform-expressing tumors among (**B**) adenoma and (**C**) squamous metaplasia. (**D**,**E**) Percentage of MUC1-positive cells as compared to all epithelial cells in (**D**) adenoma tumors and (**E**) squamous metaplasia tumors. Each dot represents one tumor. Horizontal bar represents median. For statistical analysis, the Steel–Dwass test was performed. *** *p* < 0.001.

**Figure 3 cancers-15-01920-f003:**
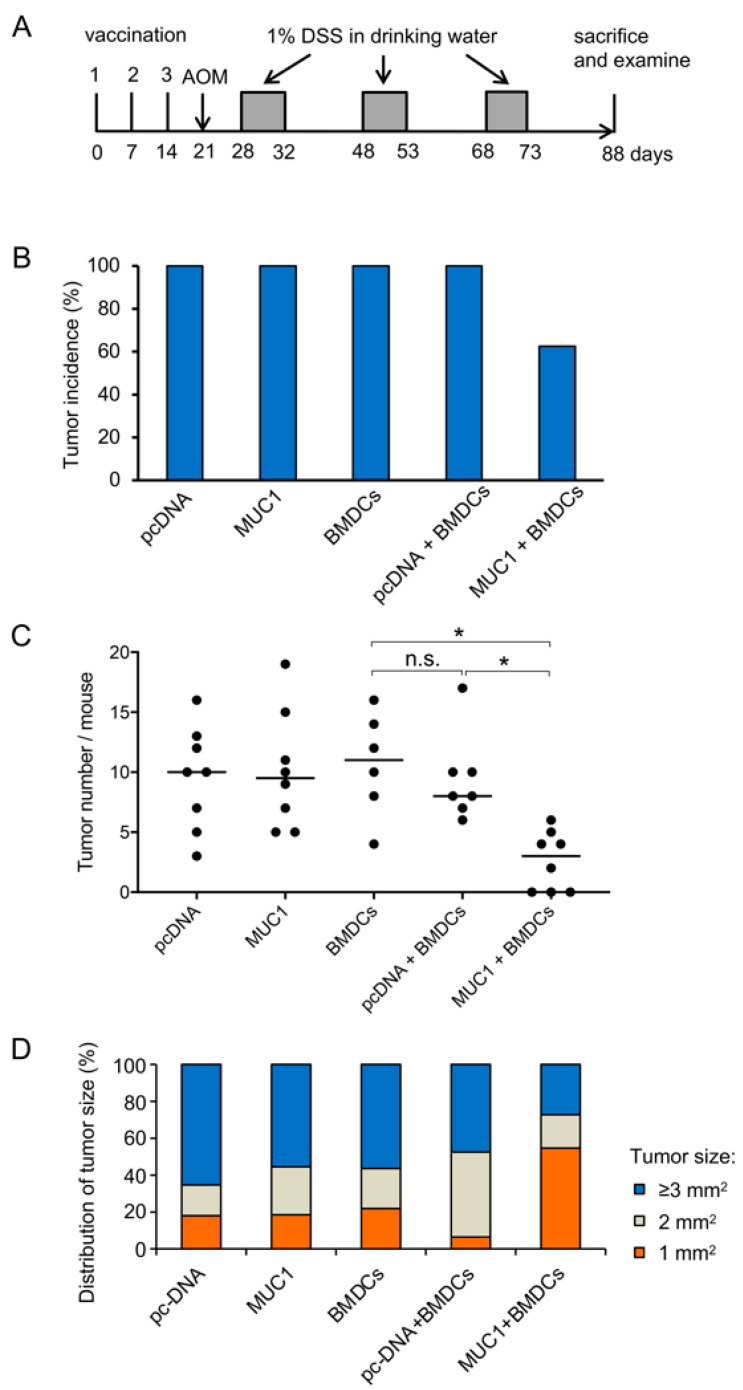
Efficacy evaluation of vaccination in MUC1.Tg mice after azoxymethane-dextran sulfate sodium (AOM-DSS) treatment. (**A**) Experimental protocol. MUC1.Tg mice were divided into five groups and vaccinated with the following: (1) mock plasmid (pcDNA), (2) *MUC1* plasmid (MUC1), (3) BMDCs (BMDCs), (4) pcDNA mixed with BMDCs (pcDNA + BMDCs), and (5) MUC1 + BMDCs, each followed by AOM-DSS treatment. Mice received three vaccinations at weekly intervals in the footpad with 100 μg of *MUC1* DNA dissolved in 50 μL of Hanks’ Balanced Salt Solution mixed with or without 5 × 10^5^ BMDCs per mouse, per vaccination. One week following the third vaccination, azoxymethane (AOM) was injected at 10 mg/kg body weight in the intraperitoneal site, followed by periodical administration of 1% (*w*/*v*) dextran sulfate sodium (DSS) in drinking water. Mice were sacrificed two weeks after the last DSS administration. (**B**) Tumor incidence in AOM-DSS-treated MUC1.Tg mice after vaccination, all groups are *n* = 10. (**C**) Number of tumors per mouse in vaccinated MUC1.Tg mice. Each dot represents one mouse. The horizontal bar represents median. (**D**) Distribution of tumor size in vaccinated mice. MUC1.Tg mice vaccinated with MUC1 + BMDCs showed a higher percentage of small (1 mm^2^) tumors and a lower percentage of large (≥3 mm^2^) tumors compared to all other groups. For statistical analysis in (**C**), the Steel–Dwass test was performed. * *p* < 0.05, n.s.: not significant.

**Figure 4 cancers-15-01920-f004:**
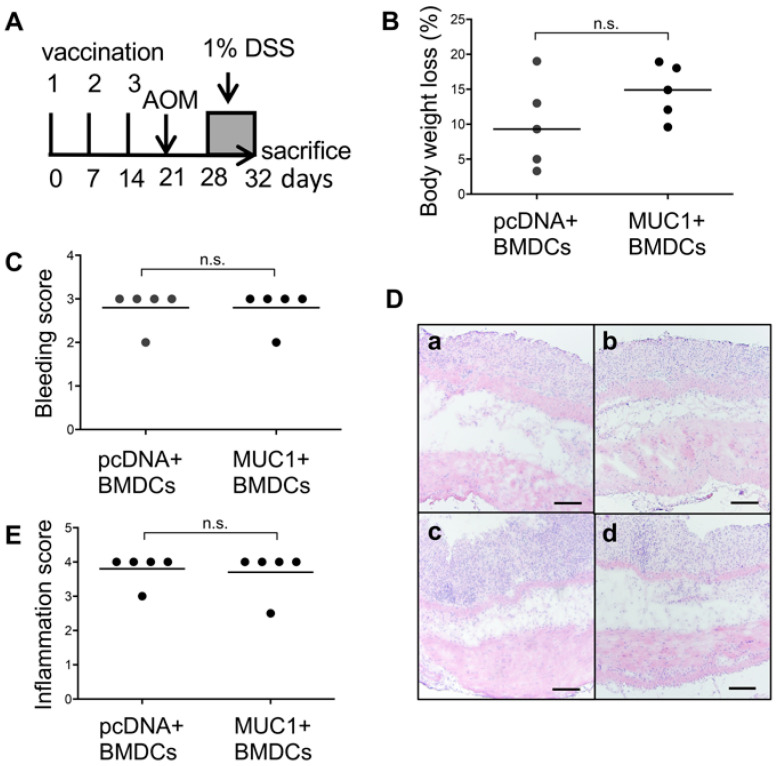
Effect of vaccination with MUC1 + BMDCs on the degree of colonic inflammation induced by azoxymethane-dextran sulfate sodium (AOM-DSS) treatment in MUC1.Tg mice. Colonic inflammation was evaluated in pcDNA + BMDCs and MUC1 + BMDCs vaccinated mice, after injection of AOM (10 mg/kg) and 5 days of 1% DSS administration. (**A**) Experimental setup. (**B**) Body weight loss. (**C**) Bleeding score. (**D**) Representative images of Hematoxylin/Eosin-stained cryosections of the colorectal (**a**,**c**) and anal (**b**,**d**) region in pcDNA + BMDCs (top) and MUC1 + BMDCs (bottom) vaccinated mice. Scale bar shows 100 μm. (**E**) Inflammation score. In (**B**,**C**), and (**E**), each dot represents one mouse. The horizontal bar represents median. Mann–Whitney U test. n.s.: not significant.

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
