# Peer review of "Prevention of Inflammation-Driven Colon Carcinogenesis in Human MUC1 Transgenic Mice by Vaccination with MUC1 DNA and Dendritic Cells"

_cancers, 2023, doi:10.3390/cancers15061920_

Round 1

Reviewer 1 Report

The manuscript reported investigations conducted by Irimura group is a very elegant study that tackles the use of a DNA vaccine targeted at human MUC1 and the use of bone-marrow dendritic cells (BMDCs)- vaccines.

Overall the study is very well comprehensive and clear. The novelty seem to be on the combination of DNA and DC vaccines

I have the following concerns:

-        It needs to be clarified why BMDC were used with no antigen loaded. What are they presenting, and what is their maturation level?

-        And what is the role of BMDC since figure 3, apparently BMDC alone substantially increases mortality.

-        Is it possible to prove BMDC were infiltrated into tissues after administration?

Reviewer 2 Report

A de novo colon cancer model that is induced by AOM-DSS was used to mimic human cancer development and to show that prior vaccination could effectively reduce tumor development. Three rounds of MUC1 vaccination was performed before AOM intraperitoneal injection which was followed by cycles of DSS in drinking water. The authors state they wished to determine if BMDC which are antigen presenting cells could enhance the efficacy of the DNA MUC-1 vaccine by behaving as adjuvants. The model they chose fits their statement. The figures are clear and the experimental plan for each was shown clearly.  However, I have several questions about their work.

11)  Would it not have been better to load the DC with the antigen first before using the DC as vaccines? What percentage of DC took up the MUC1 plasmid DNA in this model of co-injection? The authors refer to the work of other groups on this point but did not produce any studies of their own to support their point.

22) How certain are the authors that there was no LPS contamination of the plasmid DNA and that their results may not also be due to LPS activation of TLR4 pathways? Was LPS contamination evaluated in plasmid preparations?

33)  What quality controls were in place for ensuring the DC generated from the BMDC were consistently the same for each vaccination? Was the viability and surface marker expression confirmed prior to vaccination?

44) Why 3 rounds of MUC1 vaccination? What was the basis for selecting this schedule? Was there previous work showing optimal immune induction with this strategy?  If so what were the parameters that were measured?

  5) MUC1 transgenic mice express human MUC1 under its own promoter, but MUC1 expression was not observed in normal tissue in these mice though it is supposed to be expressed in normal human tissue. Is this because expression levels are so low that they could not be detected by immunohistochemistry? Or is the expression pattern dissimilar to human tissue?

6 6) Any observation of differences in immune cell infiltration in the tumors after therapy? The authors mentioned immune cells were evaluated in the inflammation model, why not the tumor model as well?

7 7) What about T cells recognizing tumor antigen MUC-1? Was any attempt made to evaluate these especially in the mice that developed tumors in the MUC1 DNA and BMDC group? The authors list a previous publication from their group showing that CD4 T cells were necessary for anti-tumor response of tumor cells expressing MUC1 after vaccination. So, they assumed the same here. Why did the authors not choose to examine Treg cells and CD4 T cells, given their previous work. The manuscript would be better if they had performed some immune analysis as well.

   8) Why did not all the mice receive protection from tumor development? Is this a failing of the vaccination attempt or the development of alternative strategies of immune evasion? Please discuss this in the discussion section.

  9) Have they protected the mice by vaccination or only delayed the onset of tumors? Would there be more, large tumors if they waited longer than 2 weeks after the last DSS dosing in the MUC1 and BMDC treated group? Please explain why this is not a likely scenario.

   10)  IS it possible to examine  MUC1 expression in the tumors to see if MUC1 expression was reduced.

Reviewer 3 Report

                As the authors state, this paper is the first to report the efficacy of a vaccine in preventing the progression of an experimentally induced colorectal cancer in an animal model. The mechanisms by which this prevention is achieved are not described in this paper.

            The authors have already made previous contributions in relation to the MUC1 transgenic mouse model.

            This paper is excellently written. Methods and Results are well described. Discussion is well planned and Conclusions are supported by the observations made.

As each Figure should be understandable by itself,  the legend of Figure 4 should include:   Each dot represents one mouse. The horizontal bar represents median.

Round 2

Reviewer 2 Report

Thank you for your reply to my queries  and the additions to the manuscript.